# Effect of Rhizome Severing on Survival and Growth of Rhizomatous Herb *Phragmites communis* Is Regulated by Sand Burial Depth

**DOI:** 10.3390/plants11233191

**Published:** 2022-11-22

**Authors:** Shanshan Zhai, Jianqiang Qian, Qun Ma, Zhimin Liu, Chaoqun Ba, Zhiming Xin, Liang Tian, Lu Zong, Wei Liang, Jinlei Zhu

**Affiliations:** 1Institute of Applied Ecology, Chinese Academy of Sciences, Shenyang 110016, China; 2College of Resources and Environment, University of Chinese Academy of Sciences, Beijing 100049, China; 3College of Forestry, Henan Agricultural University, Zhengzhou 450002, China; 4Experimental Center of Desert Forestry, Chinese Academy of Forestry, Dengkou 015200, China; 5Institute of Ecological Conservation and Restoration, Chinese Academy of Forestry, Beijing 100093, China; 6Institute of Landscape and Plant Ecology, Faculty of Agriculture, University of Hohenheim, 70599 Stuttgart, Germany

**Keywords:** arid sand dunes, adaptive strategy, clonal plants, population regeneration, sand burial, vegetation restoration

## Abstract

Rhizome fragmentation and sand burial are common phenomena in rhizomatous clonal plants. These traits serve as an adaptive strategy for survival in stressful environments. Thus far, some studies have been carried out on the effects of rhizome fragmentation and sand burial, but how the interaction between rhizome fragmentation and sand burial affects the growth and reproduction of rhizomatous clonal plants is unclear. We investigated the effect of the burial depth and rhizome fragment size on the survival and growth of the rhizomatous herb *Phragmites communis* using 288 clonal fragments (6 burial depths × 8 clonal fragment sizes × 6 replicates) in a field rhizome severing experiment. The ramet survival of the rhizomatous species significantly increased with the sand burial depth and clonal fragment size (*p* < 0.01), and the effects of the clonal fragment size on ramet survival depended on the sand burial depth. Sand burial enhanced both the vertical and horizontal biomass (*p* < 0.05), while the clonal fragment size affected the vertical biomass rather than the horizontal biomass. Sand burial facilitated the vertical growth of ramets (*p* < 0.05) while the number of newly produced ramets firstly increased and then decreased with the increasing clonal fragment size, and the maximal value appeared in four clonal fragments under a heavy sand burial depth. There is an interaction between the burial depth and rhizome fragment size in the growth of rhizome herbaceous plants. The population growth increases in the increase of sand burial depth, and reaches the maximum under severe sand burial and moderate rhizome fragmentation.

## 1. Introduction

Clonal plants produce offspring through vegetative propagation and dominate various terrestrial ecosystems globally [1,2]. Important aspects that vary across clonal species are the persistence of clonal growth organs connecting ramets and the capacity of physiological integration, which have demographic and ecological consequences [3]. Previous studies suggested that clonal species in infertile and frequently disturbed environments tend to maintain a persistent connection between ramets and exhibit high levels of physiological integration compared with those in fertile and productive environments [4,5]. Additionally, the persistent connections permit extensive physiological integration that allows ramets to relieve the negative effects of heterogeneous and stressful environments [3,6,7].

In natural ecosystems, the persistent rhizomes of clonal species have multiple functions, including resource storage, regulation of heterogeneous resource distribution, maintenance of bud banks and promotion of vegetation restoration under environmental stresses and following disturbances [8,9,10]. In highly disturbed habitats, clonal plants are often broken into small fragments of various sizes and are buried at various soil depths [11,12]. Accordingly, experimental severing has long been used to assess the importance of physiological connection and integration between ramets and evaluate the consequences of disrupted integration on ramet survival, growth, and offspring ramet production as well as population expansion via rhizome systems [9,13,14]. Studies indicate that disrupted connections often have negative effects on ramets [15,16,17]. Because the rhizome severing interrupts the flow of resources among ramets, the ramet survival and biomass accumulation following severing strongly depend on the level of stored resources [18], which is closely related to the length of the rhizome system and the ability to provide water and soil nutrients [4,19,20]. Thus, the clonal fragment size caused by different severing positions might be the key factor affecting plant performance following severing due to the level of reserves stored in clonal fragments [21]. For instance, an increasing clonal fragment size (e.g., the diameter of the rhizome) significantly increases the fragment survival of clonal shrubs [12]. Large rhizome or stolon sizes may positively affect the amount of resources stored in them and, in turn, increase the survival and regrowth of clonal fragments [22]. Despite the substantial number of studies on the consequences of rhizome severing on clonal species, how rhizome severing induces clonal fragment size interaction with environmental conditions and disturbance to affect clonal performance has been rarely explored.

In sand dune ecosystems, aeolian activity acts as a strong selective force and greatly influences individual growth, population maintenance, and colonization as well as vegetation restoration or recovery [23]. The strong aeolian activity in sand dunes often leads to changes in the sand burial depth. Sand burial is a predominant disturbance for plant species in arid and semi-arid sand dunes, and the biotic (e.g., pathogen activity) [23], and abiotic (e.g., light, temperature and moisture) conditions dramatically change with the sand burial depth [24,25]. The sand burial depth can directly affect plant survival and growth by changing below-ground plant structures [14,26,27]. It has been proved that rhizomatous plants could adjust their biomass allocation and the relationship between horizontal rhizome extension and vertical ramet growth in response to the changes in sand burial depth under wind erosion [28].

In this study, we selected the typical rhizomatous herbaceous species in semi-arid and arid sand dunes of northern China, *Phragmites communis*, as the target species, and examined the effects of the clonal fragment size (by different experimental severing positions), sand burial depth, and their interaction. *P. communis* is one of the pioneer colonists on active sand dunes in Inner Mongolia, and is often selected for mobile sand fixation and vegetation restoration [29,30]. It can reproduce through both vegetative propagation by belowground rhizomes and sexual propagation via seeds, but few seedlings are found in the field. Moreover, this species can inhabit various dune positions with different sand burial depths [29]. We experimentally severed the rhizomes at different positions with different numbers of ramets remaining on them and investigated the survival of ramets, the biomass allocation (vertical biomass and horizontal biomass), the production of new ramets, the increment in ramet height, the number of new rhizomes, and the increment in the rhizome length of different clonal fragment sizes under different sand burial depths. This study aimed to answer the two following questions: (1) How do the sand burial depth and clonal fragment size affect the survival, growth and population expansion potential of rhizomatous species? (2) Doed an interactive effect between the sand burial depth and clonal fragment size exists?

## 2. Results

### 2.1. Ramet Survival of Various Clonal Fragment Sizes under Different sand Burial Depths

There was an interactive effect between the sand burial depth and clonal fragment size on ramet survival (*p* < 0.05, Table 1). Ramet survival increased linearly with the increasing sand burial depth (*p* < 0.01), and it increased significantly with the increasing clonal fragment size (i.e., with different numbers of ramets remaining) (*p* < 0.01) (Table 1, Figure 1).

### 2.2. Effects of Sand Burial Depth and Clonal Fragment Size on Biomass Pattern

There were no significant interactive effects of the sand burial depth and clonal fragment size on both the vertical and horizontal biomass (Table 1). Both the vertical- and horizontal biomass significantly increased with the increasing sand burial depth (*p* < 0.05), whereas the clonal fragment size had contrasting influences on the vertical biomass and horizontal biomass, i.e., the vertical biomass decreased with the clonal fragment size while there were no significant differences in the horizontal biomass among different clonal fragment sizes (Figure 2A–D).

### 2.3. Effects of Sand Burial Depth and Clonal Fragment Size on Clonal Traits

The sand burial depth and clonal fragment size interactively influenced the ramet height (*p* < 0.01), but with no interaction in the ramet number (Table 1). There were no significant differences in the number of newly produced ramets (i.e., the increment in ramet number) among the different sand burial depths (Figure 3A), whereas the increment in ramet height under heavy sand burial depths (40–60 cm and 60–80 cm) was significantly higher than that under other burial depths (*p* < 0.05) (Figure 3C). With the increasing clonal fragment size, the number of newly produced ramets firstly increased and then decreased and the clonal fragment with four ramets remining—which is medium fragmentation—produced more new rhizomes (Figure 3B), but the increments in ramet height were similar among the different clonal fragment sizes, except for those of no ramets and only one ramet remaining, which were significantly lower (*p* < 0.05) (Figure 3D).

In contrast to the effects on ramet number and height, the sand burial depth and clonal fragment size interactively influenced the number of newly produced rhizomes (*p* < 0.01), while there was no interaction between the sand burial depth and clonal fragment size in the increment in rhizome length (Table 1). The number of newly produced rhizomes significantly increased with the increasing sand burial depth (*p* < 0.05) (Figure 3E), while it firstly increased and then decreased with the increasing clonal fragment size, and the clonal fragment with five ramets remaining produced more new rhizomes (Figure 3F). The increment in rhizome length gradually increased with the increasing sand burial depth (Figure 3H), but it fluctuated among different clonal fragment sizes and did not show an obvious trend (Figure 3G).

## 3. Discussion

### 3.1. Effects of Clonal Fragment Size and Sand Burial Depth on Ramet Survival

Under the harsh conditions in arid sand dunes, the physiological integration via clonal growth organs (i.e., rhizome) guarantees the resource supply and sharing between ramets, and endows clonal plants with specific adaptive strategies for sandy environments, allowing them to dominate in sand dunes [23,27,31]. However, due to serious wind erosion, some parts of belowground rhizomes are exposed to the soil surface and lose vigor, and sometimes also being disconnected due to mechanical damage, with subsequent loss of the resource-sharing function [12]. Under these circumstances, ramet survival and population persistence greatly rely on the resource storage (e.g., water and carbohydrate) in belowground rhizome systems and the soil water and nutrient uptake abilities they possess [4,19,20]. In this study, the ramet survival percentage increased with the clonal fragment size, which is consistent with the finding that the survival rate of rhizome fragments increases with the rhizome diameter [12], indicating that, with an increasing clonal fragment size, more resources can be stored and the survival of the attached ramets can be ensured.

The increasing survival percentage of ramets with the sand burial depth is mainly attributed to the increasing soil moisture content in deep sand profiles. It has been found that unlike with other ecosystems, the soil moisture content increases with the sand burial depth in sand dunes [32]. Additionally, the increased soil moisture in deep sand profiles facilitates ramet survival in this seriously drought stressed environment. Moreover, the interactive effects of the clonal fragment size and sand burial depth on ramet survival indicate that the effects of the clonal fragment size on ramet survival greatly depend on the sand burial depth. It has also been found that sand burial is one of the essential prerequisites for the clonal fragment survival of clonal shrub species [12]. The clonal fragment size represents the level of resource storage that ramet survival needs, while the increased soil moisture content at a heavy sand burial depth provides the external water availability for ramet survival. Thus, the clonal fragment size and sand burial depth seem to be the important biotic and abiotic factors determining ramet survival.

### 3.2. Effects of Sand Burial Depth and Clonal Fragment Size on Biomass Pattern and Ramet Production and Growth

In this study, the sand burial depth and clonal fragment size had additive influences on the biomass pattern of *P. communis*. Both the vertical and horizontal biomass significantly increased with the sand burial depth, which means that, under favorable water conditions in deep sand profiles, the clonal fragments grow much better. This result suggests that the effects of sand burial and wind erosion on the biomass allocation pattern of clonal species are different. For *P. communis*, while the total biomass does not differ among wind erosion depths, the horizontal and vertical biomasses, respectively, decrease and increase significantly with the increasing wind erosion depth. Therefore, there is a trade-off between the horizontal biomass and vertical biomass [28,29]. The clonal fragment size affects the biomass allocation pattern of the target species; specifically, the vertical biomass of clonal fragments with two ramets remaining is significantly higher, while the horizontal biomass does not change among different clonal fragment sizes.

The biomass allocation pattern and changes in the vertical biomass, to some extent, determine the production and growth of new ramets. The increased vertical biomass with the sand burial depth significantly increases the ramet height while having no influence on the production of new ramets. The higher increment in ramet height under heavy sand burial is consistent with previous studies which found that species tend to increase their vertical growth in response to sand burial [33]. However, the negligible influence of sand burial on ramet production in this study seems to be in contrast to the previous finding that a deeper burial markedly reduces the ramet emergence from rhizome fragments [34]. This further implies that the increased vertical biomass a under deeper sand burial supports both vertical growth and new ramet production.

A previous study pointed out that rhizome severing could affect the new shoot production of perennial rhizomatous species [9], and the emergence rate of shoots increases markedly with the rhizome fragment size [12]. In this study, as regards the clonal fragment size caused by different severing positions, the number of newly produced ramets firstly increased and then decreased with the increasing clonal fragment size, which is similar to the changes in the vertical biomass with the clonal fragment size. This implies that the vertical biomass reflects the resource supply level and determines the production of new ramets on clonal fragments. Meanwhile, the increment in ramet height was consistent among the different clonal fragment sizes, which means that various clonal fragments tend to produce new ramets with similar sizes since the carbon costs of maintaining longerrhizomes are greater and thus might limit the ramet size [35,36]. Notably, clonal fragments with few (zero or one) remaining ramets exhibited a low number of newly produced ramets and a small increment in ramet height. This is consistent with the previous findings that the adverse effects of rhizome severing are greater for clonal fragments with fewer subtending rhizome segments or ramets [37,38]. In sum, our results suggest that the sand burial depth determines the vertical growth (i.e., the increment in ramet height) while the vertical biomass determines the production of new ramets (i.e., the increment in ramet number).

### 3.3. Effects of Sand Burial Depth and Clonal Fragment Size on the Population Expansion Potential

It has been shown that horizontal rhizomes are responsible for the spread of clones into new spaces and the recruitment of new ramets [10,39,40]. In this study, we considered the increment in rhizome number and length as the population expansion potential, since the rhizomes serve as the key belowground organs for the lateral spread and vegetative reproduction of *P. communis*. We found that both the production and length of rhizomes increased significantly with the sand burial depth. This might be due to the followng: (1) The soil moisture content increases with the sand burial depth, leading to an increase in the horizontal biomass under the favorable water conditions in deep sand profiles, promoting the production of new rhizomes, (2) Under a heavy sand burial depth, rhizomatous species tend to adopt a lateral spread via their rhizomes; under shallow sand burial depths, however, both the number of newly produced rhizomes and the increment in rhizome length are significantly lower. This is consistent with our previous study, which showed that the rhizome number and total length of *P. communis* decreased significantly under shallow sand burial depths (i.e., due to wind erosion) [28]. These findings suggest that a reduction in the burial depth of rhizomes may impede natural population regeneration and colonization [29].

Even though the clonal fragment size represents the resource supply level of population expansion, we did not find an obvious trend in the rhizome production and length increment with the increasing clonal fragment size. Specifically, the number of newly produced rhizomes firstly increased and then decreased with the clonal fragment size, but the increment in rhizome length fluctuated among the different clonal fragment sizes. This indicates that the effects of the clonal fragment size on the population expansion potential are complex, and further research is needed on the potential trade-off between the number and size of newly produced rhizomes [22,41].

It should be noted that, apart from the number of new rhizomes and the increment in rhizome length, the branching frequency and angle are also important parameters determining the space expansion of rhizomatous species [42]. Thus, the different aspects such as the belowground bud bank size and composition (as the basis of vegetative reproduction and population colonization), as well as the location and production timing of new ramets, should be systematically discussed in future studies to evaluate the population expansion potential of rhizomatous species in sand dunes [43,44,45].

## 4. Materials and Methods

### 4.1. Study Site

This study was conducted in the sand dunes near the Experimental Center of Desert Forest (106°43′ E, 40°24′ N, 1050 m a.s.l.), Chinese Academy of Forestry, Inner Mongolia, China. This region belongs to the temperate continental monsoon climate. The mean annual temperature is 7.4 °C, and the mean annual precipitation is 114 mm, of which 70% occurs in June, July, and August. The average annual potential evaporation is 2372 mm, 20.8 times the precipitation.

The area has large and dense reticulate dune chains composed of loose and impoverished mobile sand with a typical soil moisture content of 3–4%. The land forms are mainly active sand dunes, semi-fixed sand dunes, and flat sandy land. The soil types are mainly aeolian sandy soil and gray-brown desert soil. The vegetation coverage of the active sand dunes is less than 5%. Additionally, the vegetation comprises only pioneering species such as *Phragmites communis*, *Psammochloa villosa*, *Nitraria tangutorum*, and *Calligonum mongolicum*. Through long-term adaptation and evolution to a sandy environment, different psammophytes occupy their specific positions on sand dunes in response to aeolian disturbance and environmental stress.

### 4.2. Target Species

*P. communis* is a perennial rhizomatous herb species inhabiting various habitats. It is one of the pioneering species on active sand dunes in northeastern Inner Mongolia, China, and plays an important role in sand fixation and vegetation restoration [29]. This species can reproduce through sexual propagation via seeds and vegetative propagation by belowground rhizomes, but its population maintenance and colonization greatly rely on vegetative reproduction, and seedlings are rarely found in the field [29]. Per its bud types (axillary buds at shoot bases, axillary buds on rhizome nodes, and apical rhizome buds), the vegetative offspring (clonal ramets) of *P. communis* can be categorized into three types with different functions: the ramets originating from the axillary buds on rhizome nodes and apical rhizome buds are responsible for population expansion/colonization, and those sprouting from the axillary buds at shoot bases contribute more to population maintenance in situ (Figure 4) [13].

### 4.3. Experimental Design

In May (the beginning of the growing season and rhizome extension) of 2018, six active sand dune systems (including the interdune lowlands and sand dunes) were selected for the rhizome severing experiment. Clonal fragments with a similar size and no dead ramets were selected for this experiment. For each clonal fragment, we dug a trench along the extension direction of rhizomes, exposed it carefully from the sand, and then recorded the original rhizome number, ramet number, rhizome length, and ramet height. According to the vertical distribution of rhizomes, we classified the sand burial depth into three groups (six layers), i.e., light (0–10 cm, 10–20 cm), medium (20–30 cm, 30–40 cm), heavy (40–60 cm, 60–80 cm). After the original measurement, the rhizome connections between ramets were severed at different positions to produce clonal fragments of different sizes. Specifically, the rhizomes were severed to have either 0, 1, 2, 3, 4, 5, or 6 ramets remaining for the clonal fragments, and an intact rhizome system without severing was used as the control. There were 6 replicates for each treatment, and therefore 288 clonal fragments (6 burial depths × 8 clonal fragment sizes × 6 replicates) in total in this study. After the severing treatment, the removed sand was refilled carefully, and then we utilized straw checkerboard barriers to ensure that the burial depth was unchanged during the whole experiment.

### 4.4. Investigation and Sampling

Throughout the experimental period, the number of surviving ramets was monitored and recorded to calculate the survival percentage of ramets. The field sampling was conducted in October 2018, five months after the severing treatment. To keep the connections between ramets intact, all plants were carefully excavated from the sand, washed with tap water, and brought to the laboratory for measurements. The increments in rhizome number, ramet number, rhizome length, and ramet height were recorded. All these plant parts were dried at 80 °C for 48 h and weighed, and the vertical and horizontal biomasses were recorded separately. In this study, we define the ramet survival percentage as the survival index; the vertical biomass, horizontal biomass, ramet number and ramet height as the growth index; and the rhizome number and rhizome length as the population expansion index.

### 4.5. Statistical Analysis

A General linear model (GLM) was applied to test the effects of sand burial depth, clonal fragment size, and their interaction on the survival, growth, and regenerative potential of *P. communis*. One-way analysis of variance (ANOVA) was used to analyze the effects of sand burial and ramet number on ramet survival, increment in rhizome number, increment in ramet number, increment in rhizome length, increment in ramet height, vertical biomass and horizontal biomass. In all cases, significant levels of differences between means were determined using least-significant difference (LSD) tests at the 0.05 significance level. All analyses were conducted using SPSS ver. 16.0 (SPSS Inc., Chicago, IL, USA).

## 5. Conclusions

The ramet survival of rhizomatous species increases with the sand burial depth and clonal fragment size, as well as their interaction. Sand burial facilitates the growth of clonal species (both for the vertical biomass and horizontal biomass) in arid sand dunes, which might be attributed to the increased soil moisture in the deep sand profile, whereas the clonal fragment size affects the vertical biomass rather than the horizontal biomass. The sand burial depth determines the vertical growth of ramets, while the production of new ramets is greatly influenced by the resource supply level, i.e., the vertical biomass. The population expansion potential, represented by the production of new rhizomes and the increment in rhizome length, is enhanced under heavy a sand burial depth, and reaches the maximum in the combination of a heavy sand burial depth and medium fragmentation.

## Figures and Tables

**Figure 1 plants-11-03191-f001:**
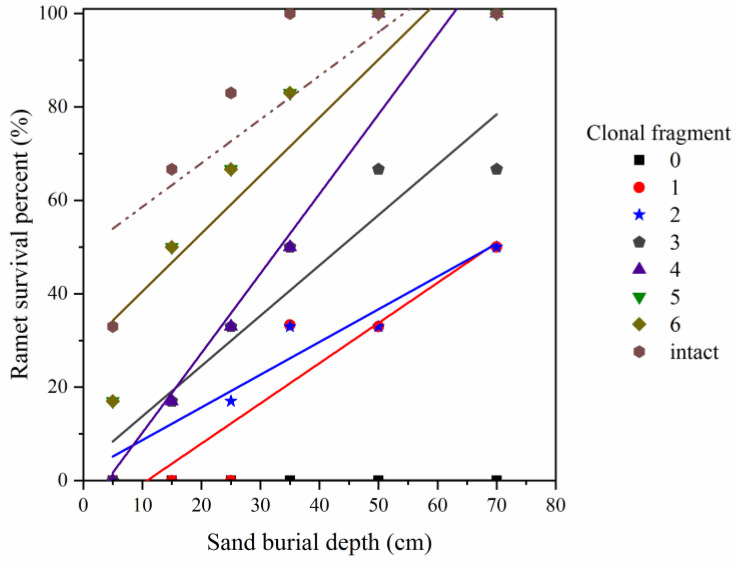
Effects of sand burial depth and clonal fragment size on ramet survival percentage. The variable y is ramet survival percentage, and the variable x is sand burial depth: y_0_ = 0, R^2^ = 1, *p* = 0.003; y_1_ = 0.86x − 9.30, R^2^ = 0.86, *p* = 0.008; y_2_ = 0.70x + 1.65, R^2^ = 0.92, *p* = 0.002; y_3_ = 1.08x + 2.98, R^2^ = 0.89, *p* = 0.005; y_4_ = 1.70x − 6.82, R^2^ = 0.93, *p* = 0.002; y_5_ = 1.24x + 28.06, R^2^ = 0.84, *p* = 0.009; y_6_ = 1.24x + 28.06, R^2^ = 0.84, *p* = 0.009; y_intact_ = 0.94x + 49.25, R^2^ = 0.69, *p* = 0.041.

**Figure 2 plants-11-03191-f002:**
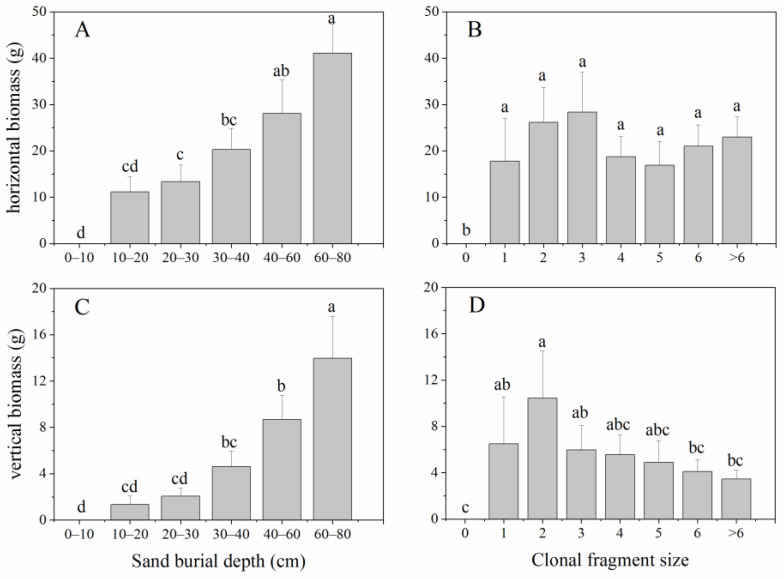
The effects of sand burial depth and clonal fragment size on the horizontal (**A**,**B**) and vertical biomass (**C**,**D**) (mean ± SE) of *P. communis*. Different letters indicate significant differences in biomass among different sand burial depths and/or clonal fragment sizes at the *p* < 0.05 level.

**Figure 3 plants-11-03191-f003:**
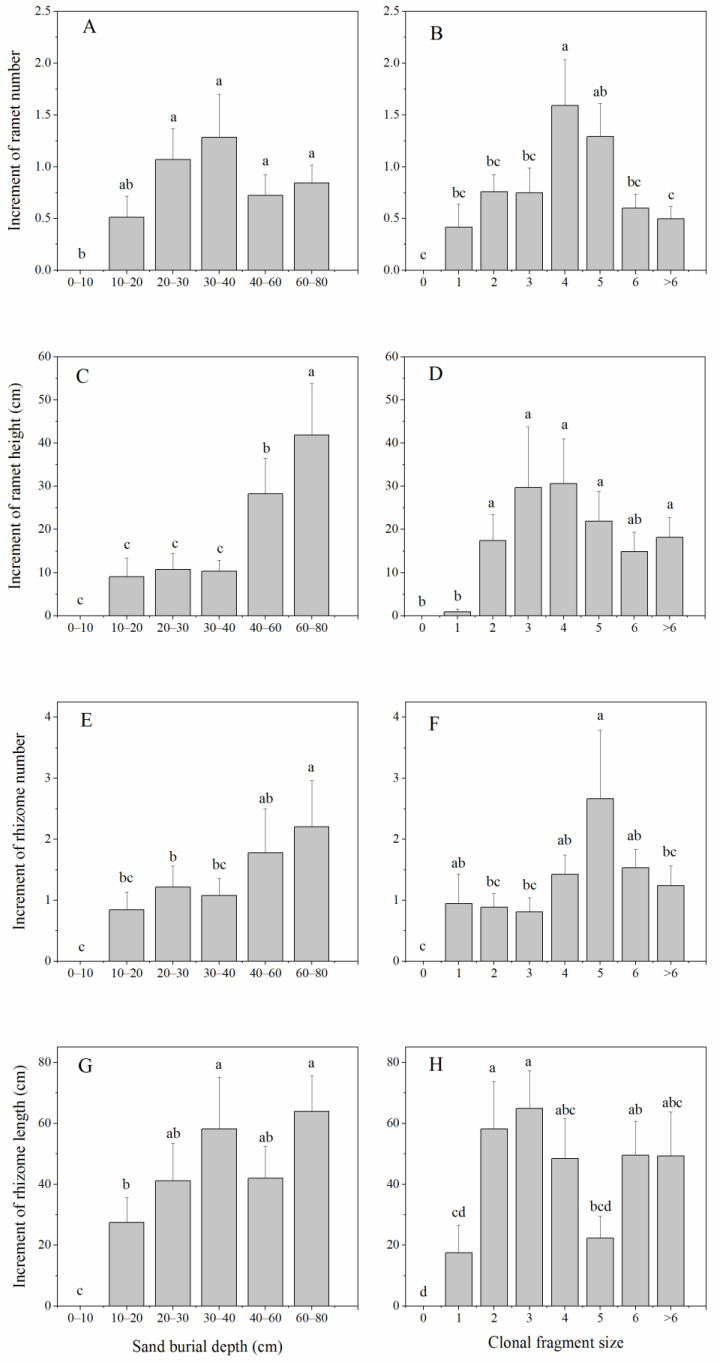
The effects of sand burial depth and clonal fragment size on the number of newly produced ramets (**A**,**B**), the increment in ramet height (**C**,**D**), the number of newly produced rhizomes (**E**,**F**) and the increment in rhizome length (**G**,**H**) of *P. communis*. Different letters indicate significant differences in biomass among different sand burial depths and/or clonal fragment sizes at the *p* < 0.05 level.

**Figure 4 plants-11-03191-f004:**
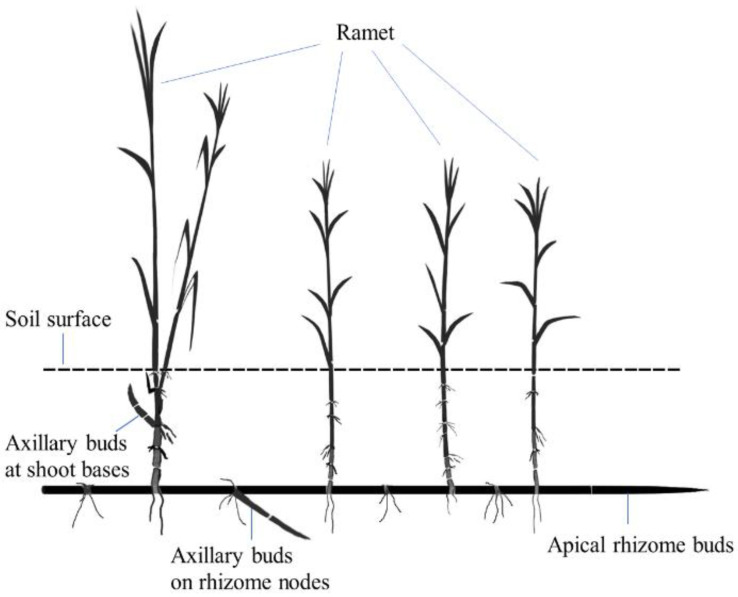
Schematic representation of belowground and aboveground organs in *Phragmites communis*.

**Table 1 plants-11-03191-t001:** Results of general linear model (GLM) on the effects of sand burial depth (SBD), clonal fragment size (CFS), and their interaction (SBD × CFS) on ramet survival and growth of *Phragmites communis*. The *p* values in bold indicate the significant differences at the 0.05 level.

Variables	SBD		CFS		SBD × CFS	
	F	*p*	F	*p*	F	*p*
Ramet survival	26.43	<0.01	21.12	<0.01	1.50	0.04
Horizontal biomass	9.69	<0.01	2.66	0.01	0.93	0.56
Vertical biomass	11.59	<0.01	2.72	<0.01	1.34	0.10
Increment in ramet number	3.69	<0.01	3.42	<0.01	0.83	0.74
Increment in ramet height	12.63	<0.01	5.28	<0.01	1.91	<0.01
Increment in rhizome number	6.07	<0.01	3.37	<0.01	2.14	<0.01
Increment in rhizome length	5.92	<0.01	4.31	<0.01	1.01	0.46

Note: the degrees of freedom for the effects of sand burial depth are 5288; the degrees of freedom for the effects of clonal fragment size are 7288; the degrees of freedom for their interaction are 35,288.

## Data Availability

The data is contained within the article.

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
