# Peer review of "Effect of Rhizome Severing on Survival and Growth of Rhizomatous Herb Phragmites communis Is Regulated by Sand Burial Depth"

_plants, 2022, doi:10.3390/plants11233191_

Round 1

Reviewer 1 Report

Dear Authors,

My review of the manuscript of “Effect of rhizome severing on survival and growth of rhizomatous herb Phragmites communis is regulated by sand burial depth (Manuscript ID:) is as follows:

The manuscript aims to demonstrate the clonal growth of arid environment. In this habitat, clonal growth is often affected by burial with sand, about which we have little knowledge so far. The manuscript shades our knowledge of clonal growth and influencing factors, so I think it is interesting. The manuscript investigated the effect of wind burial on the clonal architecture of P. communis. According to the results of the study, rhizomes have a positive effect if they are located in deeper. This can be explained by the fact that a thicker sandlayer protects them from drying out, fragmentation, and degradation in an arid environment. However, I think there is definitely a depth where the effect is already negative. What is this depth range where the negative effect already starts and prevails? It would be interesting to investigate this as well. (And also how likely is it that a clone or clone fragment will be in this depth ranges?) The manuscript is basically well written. The number of errors is not many, I have marked them, but I recommend an English language check, because the text is a bit stuttering. Please refer to the description and characteristics of the area in the Materials and Methods. If there is already a paragraph in the Discussion which begins with "In conclusion", then it is worth treating this as a separate section as Conclusion section. The reference list does not fully correspond to the format expected in the paper, it is advisable to correct this as well.

Overall, I consider the manuscript worthy of publication after minor corrections.

Minor errors and comments:

L 15: The first sentence of the abstract is very strange. Please, rethink and rephrase it.

L 84-85: “sand fixing” or “sand fixation”?

L 105-106, 125, 152 and 301-302: the scientific name of the species should be written in italics.

L 131, 258: double space

L 259: Similar results were observed in the case of herbicidal treatment of Asclepias syriaca L. in sand grasslands (Bakacsy, L., & Bagi, I. (2020). Survival and regeneration ability of clonal common milkweed (Asclepias syriaca L.) after a single herbicide treatment in natural open sand grasslands. Scientific Reports, 10(1), 1-10.). Please cite this here.

Author Response

Point 1: The manuscript aims to demonstrate the clonal growth of arid environment. In this habitat, clonal growth is often affected by burial with sand, about which we have little knowledge so far. The manuscript shades our knowledge of clonal growth and influencing factors, so I think it is interesting. The manuscript investigated the effect of wind burial on the clonal architecture of P. communis. According to the results of the study, rhizomes have a positive effect if they are located in deeper. This can be explained by the fact that a thicker sand layer protects them from drying out, fragmentation, and degradation in an arid environment. However, I think there is definitely a depth where the effect is already negative. What is this depth range where the negative effect already starts and prevails? It would be interesting to investigate this as well. (And also how likely is it that a clone or clone fragment will be in this depth ranges?) The manuscript is basically well written. The number of errors is not many, I have marked them, but I recommend an English language check, because the text is a bit stuttering. Please refer to the description and characteristics of the area in the Materials and Methods. If there is already a paragraph in the Discussion which begins with "In conclusion", then it is worth treating this as a separate section as Conclusion section. The reference list does not fully correspond to the format expected in the paper, it is advisable to correct this as well.

Response 1: Thanks a lot for your comments. I carefully considered and revised your questions as below.

(1) However, I think there is definitely a depth where the effect is already negative. What is this depth range where the negative effect already starts and prevails? It would be interesting to investigate this as well. (And also how likely is it that a clone or clone fragment will be in this depth ranges?)

Yes, it is very interesting to discuss the potential depth threshold in affecting the growth of clonal fragments. However, our study was conducted in arid sand dunes with the mean annual precipitation is 114 mm and the average annual potential evaporation is 2372 mm. Under this severe water stressful environmental conditions, we found deep sand burial have positive effects on the survival and growth of rhizome fragments, since the soil moisture content increases with the increasing sand burial depth, which might be different from other ecosystems.

(2) The number of errors is not many, I have marked them, but I recommend an English language check, because the text is a bit stuttering.

Thank you very much for your editing and suggestions. We have revised our manuscript carefully and asked for the language editing service to improve our manuscript.

(3) If there is already a paragraph in the Discussion which begins with "In conclusion", then it is worth treating this as a separate section as Conclusion section

Thanks for your suggestion. We present the conclusion as a separate section in the revised version. Please see “5. Conclusions” in the revised version.

(4) The reference list does not fully correspond to the format expected in the paper, it is advisable to correct this as well.

Thank you very much for your suggestion. We have carefully checked the format of references both in the main text and in the reference list, and corrected them.

Point 2: Overall, I consider the manuscript worthy of publication after minor corrections.

Minor errors and comments:

L 15: The first sentence of the abstract is very strange. Please, rethink and rephrase it.

Response 2: Thanks for your suggestion. We have rephrased this sentence in the revised version.

Point 3: L 84-85: “sand fixing” or “sand fixation”?

Response 3: Thanks for your comment. We corrected this point by using “sand fixation” instead of “sand fixing” in the revised version.

Point 4: L 105-106, 125, 152 and 301-302: the scientific name of the species should be written in italics.

Response 4: Thanks for your comment. We corrected them in the revised version.

Point 5: L 131, 258: double space

Response 5: Corrected.

Point 6: L 259: Similar results were observed in the case of herbicidal treatment of Asclepias syriaca L. in sand grasslands (Bakacsy, L., & Bagi, I. (2020). Survival and regeneration ability of clonal common milkweed (Asclepias syriaca L.) after a single herbicide treatment in natural open sand grasslands. Scientific Reports, 10(1), 1-10.). Please cite this here.

Response 6: Thanks for your suggestion. This reference has been added in the revised version.

Reviewer 2 Report

The abstract needs some English language editing.  The first two sentences could be merged. 

Use of articles and singular and plural are sometimes incorrect but the English is generally good.

The introduction is clear, well-focused and appropriately supported by the literature

Methods, Results, and Discussion require some additional clarifications.

Figures 3 and 4. The caption references “clonal fragment size” while the label on the figure x axis references “ramet quantity” .  The Clonal fragment size is clearer and should be used in the figure label.  Alternatively, the figure label could be referenced in the caption in parentheses.

Line 196-199.  These two sentences appear to contradict one another

Line 207 This sentence is not clear as written.  Should this be “ramet emergence from…”?

Line 318-319       What method was used to stabilize soil?

Please note additional edits on the manuscript

Author Response

Point 1: The abstract needs some English language editing. The first two sentences could be merged.

Response 1: Thanks for your suggestion. We have rephrased the first two sentences in the revised version. And we edited the English language in the abstract section carefully.

Point 2: Use of articles and singular and plural are sometimes incorrect but the English is generally good.

Response 2: Thanks a lot for your comment. We revised our manuscript carefully and asked one professional organization for language editing.

Point 3: The introduction is clear, well-focused and appropriately supported by the literature

Methods, Results, and Discussion require some additional clarifications.

Response 3: Thanks for your comment. We have revised these parts in the revised version.

Point 4: Figures 3 and 4. The caption references “clonal fragment size” while the label on the figure x axis references “ramet quantity”. The Clonal fragment size is clearer and should be used in the figure label. Alternatively, the figure label could be referenced in the caption in parentheses.

Response 4: Thanks for your comment. We have revised these errors and used Clonal fragment size in the revised version.

Point 5: Line 196-199. These two sentences appear to contradict one another

Response 5: Thanks for your suggestion. To be more clear and specific, we deleted the second sentence here in the revised version.

Point 6: Line 207 This sentence is not clear as written. Should this be “ramet emergence from…”?

Response 6: Thanks for your suggestion. We have rephrased this sentence in the revised version.

Point 7: Line 318-319 What method was used to stabilize soil?

Response 7: We used straw checkerboard barriers to stabilize sand. This method is recognized as an effective sand fixation method, we added this information in the revised version.

Point 8: Please note additional edits on the manuscript

Response 8: Thank you very much for your editing. We revised our manuscript carefully according to your suggestions.
